# A Palatal Speech Bulb—A Case Study

**Constance Hardwick [1] and James Puryer [2],\*** 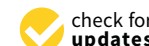

[1]  Bristol Dental Hospital, Lower Maudlin Street, Bristol BS1 2LY, UK; conniehardwick@sky.com
[2]  Bristol Dental School, Lower Maudlin Street, Bristol BS1 2LY, UK
\*  Correspondence: james.puryer@bristol.ac.uk; Tel.: +44-0117-342-4425

**Abstract:** Palatal defects of the oral cavity can be either congenital or acquired following trauma or surgical excision of malignant disease. Palatal defects can greatly affect function and subsequent quality of life. Rehabilitation using a removable obturator can be a preferable treatment option as it allows regular review post-surgery. This case study reports on the design and construction of a removable "speech bulb" obturator. A 50-year-old female patient presented complaining of nasal regurgitation and looseness of her current palatal obturator. She had previously undergone wide surgical excision of her soft palate under general anaesthesia due to adenoid cystic carcinoma. Treatment consisted of the provision of a new removable obturator, paying careful attention to the design of the "speech bulb" itself. The design of the "speech bulb" is crucial to optimise function, and the method of prosthesis fabrication is fully described. This case highlights the impact of obturator fit on a patient's quality of life and will be of benefit to clinicians from many disciplines including dentists, oral and maxillofacial surgeons, Ear, Nose & Throat (ENT) surgeons and speech and language therapists.

**Keywords:** prosthesis; obturator; rehabilitation; design; carcinoma

## 1. Introduction

Palatal defects can be either congenital, i.e., cleft palate, or acquired following trauma or surgical excision of malignant disease. Palatal defects can lead to nasal regurgitation of fluid and food, hypernasality of speech and difficulty in swallowing and whistling which can all affect a patient's physical and mental wellbeing and their quality of life [1]. The two treatment options available to occlude such defects are either surgical reconstruction (common in congenital cases), or the provision of a dental prosthesis. A removable dental prosthesis is generally preferred after malignancy or tumour removal to allow for regular review of the surgical site, facilitating identification of recurrence [2]. In addition, surgical correction may be contraindicated by systemic or local factors.

Normal physiological functions are regulated by the velopharyngeal valve which helps to separate the oral and nasal cavities during swallowing and speech [3]. This valve comprises the soft palate, lateral pharyngeal wall and posterior pharyngeal wall which directs air flow and sound into the oral and nasal cavities. When there is a defect or impairment in this mechanism, for example, after surgical excision, trauma or a congenital defect, the valve does not fully close. A dental prosthesis is designed to not only obturate the palatal defect but to restore function and to re-establish velopharyngeal valve closure [4]. A pharyngeal obturator or "speech bulb" is a removable maxillary prosthesis with an extension protruding into the pharynx. This protrusion separates the oropharynx and the nasopharynx during speaking and swallowing, aiming to improve function, speech and ultimately quality of life for the patient [5].

When designing an obturator prosthesis, it is important not to overlook fundamental principles that are applicable to all removable prostheses [6]:

- Support
- Retention
- Stability
- Aesthetics

This case study describes the design and construction of a definitive removable prosthesis for a patient following the wide surgical excision of her soft palate due to an adenoid cystic carcinoma. It highlights the importance of careful fabrication of the "speech bulb" itself to restore optimal function and improve patient quality of life.

## 2. Case Presentation Section

The presenting complaint of the female patient (CH) aged 50-years was one of nasal regurgitation and looseness of her current palatal speech bulb prosthesis which had been constructed two years previously. The patient suffered from myalgic encephalomyelitis and rheumatoid arthritis and was allergic to penicillin and non-steroidal anti-inflammatories. Her medications included paracetamol, as needed, and she was a non-smoker.

CH was diagnosed with an adenoid cystic carcinoma of her soft palate in 2012 and underwent a wide local excision under general anaesthesia. The extent of the surgical excision of the soft palate can be seen in Figure 1.

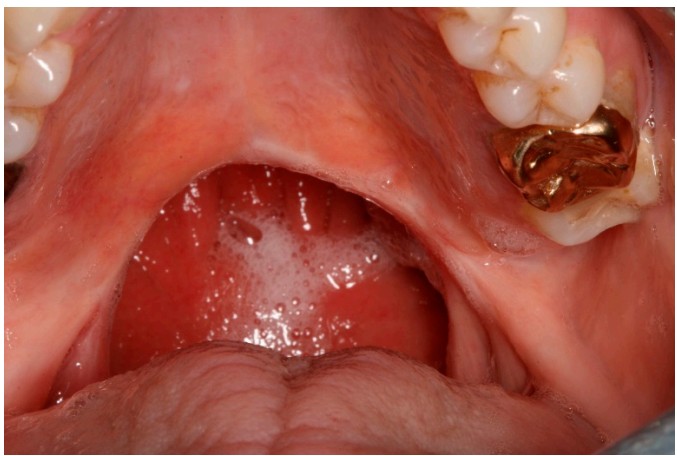

**Figure 1.** Intra-oral view showing the healed tissues following wide surgical excision of the soft palate.

A temporary palatal speech bulb was fitted at the time of initial surgery. CH underwent a further general anaesthetic approximately one month later where further impressions were taken for the construction of a second interim prosthesis. Although the surgical site had healed well, CH had struggled with this second interim prosthesis, particularly in relation to swallowing solids. Approximately two months after the initial surgery, impressions were taken for a definitive cobalt-chrome prosthesis incorporating an acrylic palatal speech bulb and this prosthesis was duly constructed and fitted. Throughout this period following surgery, CH had regular appointments with a speech and language therapist to aid her adaptation to speaking whilst wearing these various prostheses.

CH had regular reviews, and approximately 18 months after the provision of this last prosthesis, she started to struggle with retention of the prosthesis and nasal regurgitation of liquids. On re-presentation, all extra-oral and intra-oral tissues appeared healthy.

The current cobalt-chrome prosthesis had poor retention and, whilst not "dropping", contributed to the patient's nasal regurgitation of both liquids and solids due to poor posterior and lateral seals associated with the acrylic speech bulb. When the prosthesis was not worn, the patient's breathing was difficult, with obvious hypernasal speech, although she was still understandable. She was unable to

swallow any solids or liquids, including her own saliva when the prosthesis was not worn. As a result, CH was reluctant to have the prosthesis removed from her mouth for any length of time, despite its limitations. CH wore the prosthesis continually, apart from when it was removed briefly for cleaning.

CH was fully dentate in both arches, apart from the previous extractions of all four third molar teeth. There was no clinical caries seen and no periodontal pockets were found on probing, although there was evidence of plaque accumulation around the gingival margins in all quadrants.

### 2.1. Diagnoses

1   Generalised chronic gingivitis;
2   Loose upper palatal speech bulb obturator.

### 2.2. Treatment Plan

The treatment aims were to stabilise the periodontal disease, prevent further periodontal disease and to provide a more retentive palatal speech bulb prosthesis. These were achieved by:

1   Referral to a dental hygienist;
2   Provision of a new cobalt-chrome removable prosthesis incorporating an acrylic palatal speech bulb.

### 2.3. Treatment Method

CH was referred to a hygienist where oral hygiene instruction was given with respect to improving plaque control around all remaining teeth, and denture hygiene alongside dietary advice for caries prevention.

Upper and lower primary impressions were taken with an alginate impression material and an upper special tray was constructed from the resultant upper cast. An upper master impression was taken within the special tray using a medium-bodied silicone putty (Extrude, Kerr, Orange, CA, USA).

A cobalt-chrome framework was cast. In this case, the patient was fully dentate, therefore support for the prosthesis could be obtained from the upper dentition, using a number of rests within the cast cobalt-chrome framework. The options for retention however were limited due to the patient being fully dentate. Therefore, multiple occlusally-approaching clasps provided retention for the prosthesis. The framework incorporated a cast distal extension which provided support for the acrylic speech bulb. At the time of fitting, the prosthesis was both stable and retentive, and the patient reported no discomfort. A chair-side modification of the speech bulb was undertaken using cold cure acrylic resin (Tokuso Rebase, Tokuyama, Osaka Japan) to refine the lateral borders to ensure an optimal seal. The final prosthesis can be seen in Figures 2 and 3.

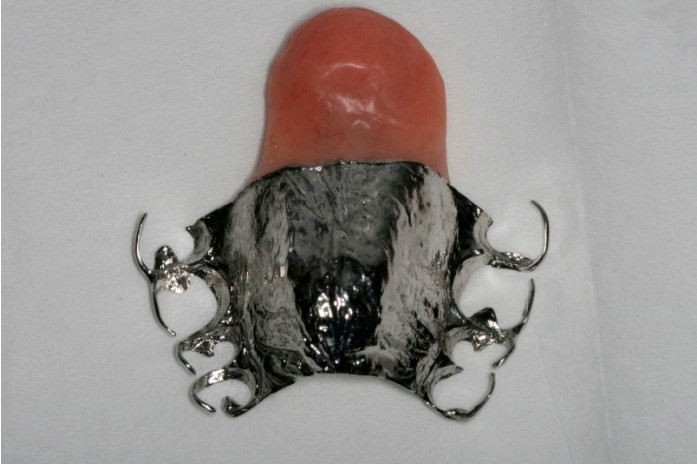

**Figure 2.** The constructed definitive prosthesis and palatal speech bulb.

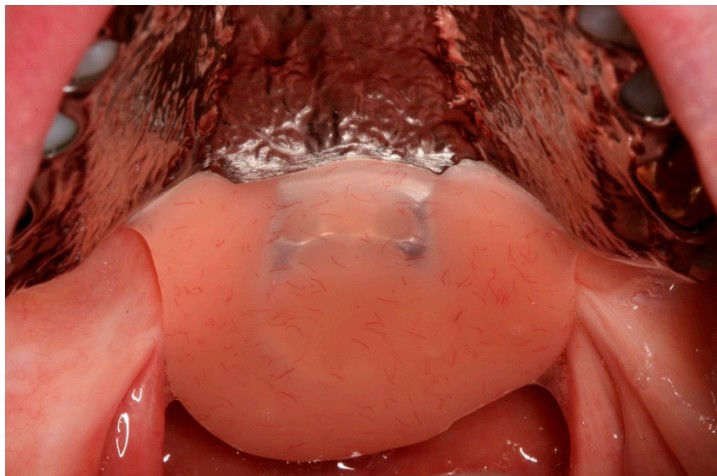

**Figure 3.** Intra-oral view showing the definitive prosthesis in-situ with good approximation of the lateral walls of the "bulb" to the soft tissues which help to provide a seal.

The patient's breathing and speech were normal, and she was able to swallow both liquids and soft solids with the prosthesis in-situ at this appointment. On removal of the prosthesis there was no evidence of food on the upper surface of the speech bulb. The patient was happy to try this new prosthesis and a review was arranged.

Three weeks later, the patient reported no discomfort, no issues with retention of the prosthesis and no issues with her speech, swallowing or nasal regurgitation. However, CH reported difficulties in breathing. The acrylic bung was adjusted to reduce the posterior extension to allow greater air flow through the oro-pharynx. This had a positive effect on her breathing, and a further review was made.

At a second review appointment two weeks later, CH reported an improvement in breathing, but was now having some issues with food debris gathering on the upper surface of the speech bulb. Further modifications to the posterior margin of the speech bulb were made using cold cure acrylic resin (Tokuso Rebase, Tokuyama, Osaka, Japan) to refine its shape. It was also noted that CH was suffering from mild denture-induced stomatitis on her hard palate. Denture hygiene was reinforced with a Miconazole oral gel prescription.

Subsequently, CH was managing well with her prosthesis and the denture-induced stomatitis had resolved. CH was placed on a three-monthly review along with further hygienist support for ongoing preventive advice.

*2.4. Ethics and Consent*

Verbal consent was obtained from the patient prior to taking the clinical photographs and no patient-identifiable data are present in this report.

**3. Discussion**

This case study describes the construction, and subsequent refinement, of a palatal speech bulb prosthesis following surgery for neoplastic disease. When an obturator is the preferred oral rehabilitation, its design is dependent upon the size and position of the defect as well as the residual anatomy that can provide support and retention for the prosthesis [7].

Defects of the soft palate can be difficult to treat, as the smallest loss of soft palate leads to altered structure and therefore function. When designing an obturator to replace tissue of the soft palate, the main goal is to restore normal velopharyngeal function [8], including: speaking, swallowing, sucking, blowing and sneezing [9]. A primary objective is to control nasal emission to prevent leakage of material into the nasal passages during swallowing, and to prevent inappropriate nasal resonance during speech [10]. Obturator design follows the basic principles for all removable prostheses of optimizing

support, stability and retention alongside maintaining good oral health. The major connector was kept free of the anterior palate to help reduce plaque accumulation around the gingival margins and allow the patient's tongue to feel some of the natural hard palate [11,12]. An alternative design could have been to construct an all-acrylic prosthesis, using Adam's cribs to gain retention. As with all prostheses, design is a compromise, with better function of the prosthesis often being traded for reduced aesthetics, comfort or oral health [13]. In this case the patient was happy to compromise aesthetics by having multiple occlusally-approaching clasps, as function and retention of the prosthesis was her main priority. This design also allowed a "fall back" position in that if one of the clasps were to fracture off, the prosthesis could most likely still be worn.

It is essential to ensure that the speech bulb itself is of the correct shape, size and position. Various techniques have been described for the fabrication of the speech bulb [2–4]. Acrylic resin was used due to its ability to be reduced or relined chair-side, as needed. In this case a chair-side addition, using cold-cure acrylic resin was carried out at the fit stage to help provide good lateral seals. Subsequent modification of the bung was undertaken to optimise function and this modification was based upon the patient's verbal feedback.

The patient's speech was very good whilst wearing the new prosthesis. Several methods of speech evaluation have been described including acoustic spectrogram [14], pressure flow technique [15] and acoustic and aerodynamic techniques [16]. If these techniques are unavailable, it has been suggested that a patient's own perception of speech can be an effective guide [17]. The involvement of a speech and language therapist, as in this case, can be very beneficial at an early stage to help perception of speech, especially when these instrumental measures are unavailable [18].

As for all patients with prostheses, oral hygiene and denture hygiene are essential. Plaque accumulation on the prosthesis can increase the risk of dental caries and periodontal disease [19] and poor denture hygiene can lead to denture stomatitis [19]. This risk of denture stomatitis is increased when patients fail to remove their prosthesis at night [20]. In this case, the risk of denture stomatitis was accepted as CH needed to wear the prosthesis full-time to restore her velopharyngeal function. The application of topical Miconazole gel along with thorough denture hygiene is proving successful.

## 4. Conclusions

This case describes the successful prosthetic rehabilitation of a patient with a soft palate defect following surgery for an adenoid cystic carcinoma. The design of the obturator followed the basic principles of removable prosthesis construction with careful consideration given to the fabrication of the speech bulb. From questioning the patient during treatment, there is no doubt that the provision of this satisfactory obturator improved her quality of life.

**Author Contributions:** J.P. treated the patient; C.H. wrote the initial draft manuscript; J.P. reviewed and edited the manuscript. Both authors have approved the final manuscript and revised the manuscript.

**Conflicts of Interest:** The authors declare no conflict of interest.

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
