# Peer review of "A Palatal Speech Bulb—A Case Study"

_reports, doi:10.3390/reports2010005_

Round 1
Reviewer 1 Report
A concise article on the construction of a palatal speech bulb obturator which is appropriate as a case report.
Line 30 - rephrase last sentence in first paragraph of introduction.
Line 33 - ...directs air pressure or air flow?
Line 136 - spelling of clinical
Author Response
Dear Reviewer Thank you for taking the time to review our paper, and for making some suggestions as to how it could be improved. We feel that we have responded to all of your comments and made minor revisions as follows: Line 30: rephrased the last sentence of the first paragraph of the Introduction Line 33: changed text to 'air flow' Line 136: changed spelling of the word 'clinical' We hope that you will find these revisions agreeable and recommend this paper for publication. Yours sincerelyReviewer 2 Report
Summary
This paper presents the case of a 50-year old female patient who had undergone a wide surgical excision of her soft palate. The case study outlines the process of fitting her with a functional and effective speech bulb. The paper is succinct and well written. The presented case adds valuable information to the field and is of interest to the journal’s readership.
Major comments
1. Abstract
a. The abstract concisely describes the main aim and outcomes from the study.
2. Introduction
a. The introduction is clear and accurate. It provides a good foundation for the case that follows.
b. Page 1 line 32-33, authors might like to split this sentence into two and define the type of physiological functions regulated by the velopharyngeal valve. Further, authors can specify that the velopharyngeal valve directs both air and sound into oral and nasal cavities.
3. Case presentation section
a. Page 4, how was the patient’s speech at the 3-week review?
4. Discussion
a. Page 5, line 168 onwards, perceptual assessment by a speech and language therapist is often the first port of call before outlined instrumental measures, which are often unavailable.
5. Conclusion
a. Line 185, was the patient’s quality of life overtly examined or questioned during treatment? This is a broad statement that could be refined.
Minor comments
1. Page 1 line 27; page 4, line 118; and page 5, line 159-160 grammatical errors.
2. There are some inconsistencies with tense throughout the paper.
3. Page 1 line 41, a reference needed.
4. Figure 1 can be referenced earlier in the text when the case is being introduced.
5. Page 3 lines 91-92, authors can specify the type of disease that is being stabilized and prevented.
6. Page 4, line 118, authors can specify that this observation was made with the prosthesis in-situ.
7. Page 5 lines 150-155, these details can be included in the earlier Case Presentation Section.
Author Response
Dear Reviewer Thank you for taking the time to review our paper and for making suggestions as to how it could be improved. We feel that we have been able to address all of your suggestions as follows: Page 1 Line 32-33: sentence has been split into two. Functions of the value have been added. Direction of air and sound has also been added. Page 4: reported on the patient's speech at the 3-week review Page 5 Lines 168 onwards: clarified the benefit of a Speech and Language Therapist at an early stage when these instrumental measures are not available. Page 5 Line 185: refined the final sentence to clarify that it was questioning the patient that informed us that her quality of life was improved. Addressed three grammatical errors where indicated. Improved the consistency of tense used throughout the paper. Page 1 Line 41: reference added Moved the reference to Figure 1 and the image earlier in the text Page 3 Lines 91-92: clarified periodontal disease Page 4 Line 118: clarified with the prosthesis in-situ Page 5 Lines 150-155: these details have been moved earlier in the Case Study and reworded appropriately We hope that you will approve of these revisions and give a favourable recommendation for publication. Yours sincerelyReviewer 3 Report
The proposed Case Report entilted "A Palatal Speech Bulb – A Case Study" describes the design of a prosthetic ‘speech bulb’ obturator and successive rehabilitation of a patient after palate surgery.
The report is exhaustive and highlight the recovery of functions in which palatal is involved.
I retain that this report is suitable for publication.
Author Response
Dear Reviewer Thank you for taking the time to review our paper. We are very pleased that you support the view that it is suitable for publication. Yours sincerely